# The Past, Present, and Future of Wheat Dwarf Virus Management—A Review

**DOI:** 10.3390/plants12203633

**Published:** 2023-10-20

**Authors:** Anne-Kathrin Pfrieme, Torsten Will, Klaus Pillen, Andreas Stahl

**Affiliations:** 1Institute for Resistance Research and Stress Tolerance, Julius Kühn Institute (JKI)—Federal Research Centre for Cultivated Plants, 06484 Quedlinburg, Germany; torsten.will@julius-kuehn.de (T.W.); andreas.stahl@julius-kuehn.de (A.S.); 2Institute of Agricultural and Nutritional Science, Plant Breeding, Martin-Luther-University Halle-Wittenberg, 06108 Halle (Saale), Germany; klaus.pillen@landw.uni-halle.de

**Keywords:** wheat dwarf virus (WDV), resistance, mastrevirus, resistance genes, Geminiviridae, resistance breeding

## Abstract

Wheat dwarf disease (WDD) is an important disease of monocotyledonous species, including economically important cereals. The causative pathogen, wheat dwarf virus (WDV), is persistently transmitted mainly by the leafhopper *Psammotettix alienus* and can lead to high yield losses. Due to climate change, the periods of vector activity increased, and the vectors have spread to new habitats, leading to an increased importance of WDV in large parts of Europe. In the light of integrated pest management, cultivation practices and the use of resistant/tolerant host plants are currently the only effective methods to control WDV. However, knowledge of the pathosystem and epidemiology of WDD is limited, and the few known sources of genetic tolerance indicate that further research is needed. Considering the economic importance of WDD and its likely increasing relevance in the coming decades, this study provides a comprehensive compilation of knowledge on the most important aspects with information on the causal virus, its vector, symptoms, host range, and control strategies. In addition, the current status of genetic and breeding efforts to control and manage this disease in wheat will be discussed, as this is crucial to effectively manage the disease under changing environmental conditions and minimize impending yield losses.

## 1. Introduction

As early as the 8th century AD, the Japanese Anthology described the first observations of viroses on *Eupatorium chinense* L., which, according to current knowledge, were caused by geminiviruses [1]. As a consequence of climate change, insect-transmitted viruses are gaining increased importance because vectors may benefit from a temperature increase in different ways [2,3,4,5]. Damage caused by viruses in agriculture includes not only yield and biomass losses but also the weakening of infected plants, making them more susceptible to abiotic and biotic stressors, so that quality losses may also occur [6]. Currently, there are no approved options for direct chemical control of viruses. So, appropriate measures in accordance with integrated pest management include farm hygiene, quarantine programs for the import and export of plant products, production of virus-free seeds and planting materials, breeding of resistant varieties, and, as a last measure, the control of vector insects by the use of chemical insecticides [7,8].

In Europe, more than 30 different viruses are known to occur in cereals [9]. These include wheat dwarf virus (WDV, family Geminiviridae, genus Mastrevirusas the causal agent of wheat dwarf disease (WDD). The virus is transmitted from plant to plant exclusively by leafhoppers [10,11,12]. The first occurrence was described in the former Czechoslovakia [10], followed by subsequent outbreaks in the 1990s [13,14,15,16]. Outbreaks vary from year to year and differ in the damage they cause, with early infections in the fall leading to drastic yield losses [17,18]. Lindblad and Waern [17] put the average yield losses in winter wheat fields at 35–90% for sites studied in Sweden, while a study in southern Finland found losses of 20–100% [18].

Due to the shift in seasons as a result of climate change and the resulting higher temperatures in late autumn and February/March [19], a longer infection period can be expected due to a higher vector activity [3], possibly leading to increased disease incidences with higher infection rates in fields. The recent increase in the incidence of WDV in European, African, and Asian cereal-growing regions is promoting research activities with regard to plant resistance in wheat and barley. This article provides an overview of the virus, its vector, and ways of control, with a particular emphasis on wheat.

## 2. Wheat Dwarf Virus (WDV)

### 2.1. Classification and Genomic Organization of WDV

WDV belongs to a group of viruses originally described as wheat dwarfing viruses within the family Geminiviridae, genus Mastrevirus [20,21,22,23].

Geminiviruses themselves are defined as plant pathogenic circular single-stranded DNA (ssDNA) viruses [24]. Their virion consists of twinned (geminate) icosahedra with a bipartite capsid [25,26] and a genome packaged in 11 subunits [1,26,27]. In addition to nanoviruses (family Nanoviridae), they are the only phytopathogenic representatives with a genome consisting of a circular ssDNA [28]. Actual research on the family Geminiviridae began in the 1980s, although they have been known since the beginning of the 20th century, mainly as causal agents of yield loss in tomato, sugar beet, cassava, maize, and cotton in tropical and subtropical countries [29,30,31,32]. Based on their genome structure, vector, host range, and phylogeny, geminiviruses are classified into 14 genera with 520 species (Figure 1) [21,22,23,33,34,35].

Currently, 45 different mastreviruses are known, which type species is Maize streak virus (MSV) [34,36], and share a common phylogenetic tree [37,38,39]. They predominantly infect monocotyledonous plants, with a few exceptions, such as *Tobacco yellow dwarf virus* [40], *Bean yellow dwarf virus* [41], and *Chickpea redleaf virus* [42,43], which can infect susceptible dicotyledonous host plants. Transmission of these viruses to host plants is mainly persistent and non-propagative through leafhoppers as vectors [42,43]. The Mastrevirus genus has a monopartite circular ssDNA genome with a length of 2.6–2.8 kb [44,45]. The genome of WDV [20], which belongs to this group, is 2.73–2.75 kb in size [14,25,46,47].

The circular genome contains two open reading frames (ORFs) on the sense side and two ORFs on the antisense side, separated by two noncoding regions that encode four viral proteins. On the virion sense strand, ORFs V1 and V2 are responsible for encoding the viral movement protein (MP) and the coat protein (CP). On the complementary sense strand are C1 and C2, which encode the replication-associated proteins (Rep, RepA) and are expressed through script splicing [48,49,50,51,52,53]. The two strands are separated by a large (LIR) and a small (SIR) non-coding intergenic unit, whose sequences are substantially involved in viral replication and regulation of gene expression [54] and control bidirectional transcription based on promoter (transcription initiation step) and terminator (transcription termination step) sequences [55,56]. Between the 5′ ends of the Rep/RepA and MP genes is the LIR sequence [57].

The replication-associated proteins (Rep, RepA) are encoded by a gene and are expressed by a complementary sense transcript. Both forms differ due to an intron in the Rep gene [16,51,58,59,60,61] and are involved in the early stages of infection [27]. Rep is involved in viral replication, while RepA affects the control of the host cell cycle to support viral replication [27]. Translation of RepA occurs directly from the native RNA transcript, whereas production of the Rep protein requires a splice cut of the RNA molecule. Therefore, the proteins have identical N-terminal sequences [62].

MP, as a product of V2, is a 10.9-kDa protein involved in systemic infection of the host by increasing the exclusion limit of plasmodesmata, allowing intercellular spread of viral DNA [63,64]. The functions of the coat protein (CP) have been studied most extensively for mastreviruses [65]. In addition to encapsulating viral DNA with a capsid, it is involved in various functions in the infection cycle, i.e., virus–vector interaction during transmission [62]. Thus, it plays an important role in vector specificity [66], viral nuclear import [67], insect transmission, systemic viral movement, and symptom development [48,65]. For the establishment of systemic infection, both MP and CP (V1 and V2) have been found to be essential, although they do not contribute to virus replication. CP binds ssDNA and dsDNA in vitro in this process, so its presence is essential for the accumulation of viral ssDNA in infected host cells and protoplasts [68].

The geminiviral transcriptional activator protein (TrAP) plays a role in pathogenicity by inhibiting a plant’s transcriptional and post-transcriptional gene silencing [69,70,71,72,73,74,75,76,77]. Enhanced viral replication is initiated by the replication enhancer protein (Ren), which interacts with host factors and Rep [66].

In several wheat isolates, a putative fifth ORF was discovered on the complementary (−) strand, coding for a protein (14.6 kDa) whose function is still unknown [14,46,47,78]. An additional ORF has not yet been detected in barley-adapted WDV isolates [79,80].

### 2.2. Life Cycle of the Virus

The life cycle of geminiviruses require both host proteins and viral proteins. Infection of the host plant begins as soon as the virus-bearing insect vector secretes saliva into the host plantit. Deposition and unpacking of the viral genome occurs in the phloem companion cells [81,82,83]. Replication of geminiviruses takes place in the nucleus of the companion cells because the sieve elements do not have a nucleus as a consequence of ontogenesis [84]. The entry of viral DNA into the nucleus is supported by the coat protein (CP). This is thought to interact with host-specific transport receptors. Within the intergenic regions, there are signal motifs controlling the two phases of replication. The onset of DNA synthesis is initiated specifically for representatives of the genus by a primer (approximately 80 bp long) located in the SIR, which is complementary to the intergenic region [49,50]. In the first phase, ssDNA is converted into a double-stranded (ds) DNA intermediate [85], which serves as a template for the production of complementary and virus-sense transcripts [55,56]. Replication of the genomic (+) DNA strand is initiated (ori) by cleavage of the virion-sense strand at a specific, highly conserved nona-nucleotide motif (5′ TAATATT ↓ AC 3′) by Rep (replication initiator protein) within the LIR sequence [57,82]. The motif is partially enclosed within the head of a stem-loop structure and contains the initiation point (↓) of the second replication phase to produce the (+) DNA strand using a rolling circle replication process [41,61,85,86,87,88].

For the amplification of viral dsDNA and the production of ssDNA genomes, the dsDNA intermediate is used as a template. Starting from the LIR, passing through the (−) and (+) strands, and continuing to the SIR, bidirectional transcription of the DNA occurs using host DNA polymerase [89]. Geminiviruses do not code for a DNA polymerase in this process, so the production of dsDNA using complementary DNA synthesis depends exclusively on host factors recruited during the early stages of replication [82]. Synthesis of the complementary minus (−) DNA strand begins at the 3′ end of a short complementary primer. This is packaged into viral particles and can hybridize with a sequence in the SIR region [85]. Transcription is bidirectional, with coding regions diverging from the LIR in both strands. For gene expression, geminiviruses use multiple overlapping transcripts [82].

The movement of the virus depends on the outcome of interaction with different parts of the cell (cytoskeleton), the type of plasmodesmata, and the ability of the virus to replicate in different cells [90]. In infected plants, electron microscopy has revealed altered nuclei in the phloem companion and in the parenchyma cells of roots and leaves [91]. In these cells, there is an accumulation of virus particles arranged in groups and rows, filling almost the entire nucleoplasm. High particle concentrations have been detected, especially in plants with wilted leaves in the stem region [92].

To spread the infection, the virus must overcome barriers such as the nuclear envelope and spread between adjacent cells [93]. Viral DNA is transported from the nucleus to the cell membrane as a V2-DNA complex with the help of the transport protein (MP), which binds to host receptors [44]. To spread the infection from one cell to another, the virus must pass through plasmodesmata. This is possible exclusively between the companion cells (CC) and the sieve element (SE) of the CC/SE complex because they are isolated from the surrounding phloem parenchyma cells, as indicated by a very low number of plasmodesmata in barley [94] and their absence in wheat [95]. Depending upon the developmental stage, the size of the protein that can pass through the plasmodesmata varies, as shown forwheat [96]. The authors furthermore demonstrated that a viral movement protein is able to increase the open width of plasmodesmata so that proteins with higher molecular weight can pass through, independent of the leaves’ developmental stage. This would facilitate the systemic movement of a virus such as WDV. WDV is distributed together with photoassimilates and other nutrients along the sieve tube with transport based on turgor-driven mass flow from source to sink [93]. For maize streak virus in maize, it has been shown that younger leaves formed after inoculation are more likely to be infected with the virus than older leaves because the viral antigen is distributed according to the age of the tissue. The virus can, therefore, be detected in the basal meristem of young leaves as it reaches them through the phloem with the metabolites of older leaves. For long-distance transport, probably only the thin-walled SEs that form the above-mentioned CC/SE complexes are relevant, while the thick-walled SEs lack CCs and, thus, the basis for virus replication [97].

Regarding the molecular mechanisms of spread and the associated interaction with host components, many questions remain open in the relationship between geminiviruses and hosts. Cell-to-cell spread is ensured by phosphorilization of the transport protein (MP) by host kinases [98,99,100]. A study of begomoviruses (Geminiviridae) in tomato (*Solanum lycopersicum*) and soybean (*Glycine max* [L.] Merr.) identified the cellular interaction partners that support the transport of the viral genome from the nucleus to the cytoplasm. For both plant species, a membrane-associated plant species–specific kinase belonging to the LRR-RLK family of proteins (leucine-rich-repeat receptor-like kinase) was discovered. Within the highly specific interaction, short-term formation of a complex of nuclear shuttle protein (NSP) and NSP-interacting kinase (NIK) occurs, which provides targeted and active recognition of nuclear pores, plasma membrane, and plasmodesmata modes. The complex presumably serves to regulate the biochemical activity of the viral protein in phosphorylating the transport protein. In this case, NSP would regulate the movement of viral DNA through the kinase activity of transmembrane receptors for this purpose. Host kinase as enzyme and viral NSP as substrate are related here [98]. Therefore, the non-host relationship between the wheat and barley strains of WDV could be due to the non-recognition of the viral protein by the plant receptor. In this case, the low incidence of winter barley infected with the wheat strain and winter wheat infected with the barley strain could be attributed to a sequence swap resulting from a mutation [101].

### 2.3. Phylogenetics

Based on phylogenetic analyses of WDV sequences from isolates of different host species, WDV has been shown to form a clade that is distinctly different from other mastreviruses and consists of multiple strains [102,103]. WDV sequence identity is below the delimitation criterion of <75% for the Mastrevirus species [36,104].

A further Mastrevirus species was later identified in *Avena fatua* in Germany, based on sequences of isolates collected from plant samples from cereal fields. *Oat dwarf virus* (ODV) is closely related to the WDV species but is distinct from wheat and barley strains and appears to be one of the causal agents of WDD in oats [104], with symptoms comparable to those of WDD (Figure 1a). Although some relationships exist between WDV and ODV based upon a sequence analysis, the whole genome of ODV has only a nucleotide sequence similarity of approx. 70% compared to the wheat and barley strains of WDV. Based on a phylogenetic analysis, a revision of the classification of the Mastrevirus species into five phylogenetic groups (A–E) was proposed in 2013. In this context, WDV strains that preferentially infect wheat (WDV-W) or barley (WDV-B) should be assigned to groups A and C, respectively [37]. Phylogenetic analysis of 230 isolates identified six strains (A–F) based on sequence similarity. Strains A- and F- were assigned to WDV-B (Figure 1, Clade A1, A1, WDV-Bar), and strains B–E were mainly assigned to WDV-W (Figure 1, Clade WDV-A, WDV-B) [105].

Macdowell et al. [14] and Matzeit [25] sequenced a 2749 bp Swedish isolate (WDV-S), which was isolated from wheat in 1969 [78]. Two other wheat-adapted isolates from the Czech Republic (WDV-C) [46] and France (WDV-F) [47] showed a genome size of 2750 bp. Sequence analyses showed that barley WDV isolates had at least 94% similarity, whereas wheat isolates had at least 98.3 to 98.8% sequence similarity with the respective strains [46,47,78]. LIR and SIR represent the most variable parts of the WDV genome [104]. Within the genomes, nucleotide exchanges in coding regions were observed but did not result in amino acid sequence substitutions, so this had no effect on the gene products [78].

Depending on the WDV isolate, differences in WDV virulence can be observed. Significantly increased symptoms of a WDV infection can be attributed to amino acid substitutions in the CP gene. This was reported in a Ukrainian study in which the Ukrainian isolate Khm-K-Ukr caused a significantly greater reduction in seeds per ear and thousand-grain weight compared to the isolate MIP-12-Ukr, which had fewer mutations in the CP gene than Khm-K-Ukr. The authors of the study suggested that the isolate MIP-K-Ukr has a higher divergence potential so that the CP sequence contains more non-synonymous changes that are subject to selection [106]. This has already been observed for the *maize streak virus*, where even a few changes in nucleotide sequence have large effects on virus functionality [107].

Within a host, different WDV populations can occur [108], and a lack of antagonism between isolates may favor recombination between viral sequences during host infection. Such a case has already been described for the isolate WDV Bar [TR]. The isolate is a variant of the barley WDV strain described in infected barley in Turkey [109]. Whole genome sequence analysis showed that the barley WDV isolate partially corresponds to a novel WDV-like Mastrevirus species [110]. In addition to the WDV Bar [TR] isolate, sequence alignment analysis of field isolates revealed regions of the viral genome with short, few-nucleotide recombination patterns between wheat and barley strains. This suggests that sequences from barley strains were replaced by functionally homologous sequences from wheat strains [108]. Moreover, intra-specific recombinant genomes were detected with two WDV wheat strains in China [111]. In this context, it should be noted that defective forms of wheat and barley strains containing at least part of the SIR and LIR sequences have also been detected in WDV-infected plants [15,108]. Putative recombinant isolates have also been identified for other members of the Mastrevirus genus, such as the maize streak virus [112].

## 3. Wheat Dwarf Disease (WDD)

### 3.1. History

The first dwarfing of wheat in Europe was observed in the early 20th century, with characteristic heavy tillering, dwarfing, and deformation of the plants and subsequent death, while the first similar symptoms were described as early as 1863 in a region that is now part of Poland [113]. In Sweden, the leafhopper species *Psammotettix alienus* was made responsible for this by Tullgren in 1918 [114] (Table 1). At that time, it was assumed that other insects besides *P. alienus* were involved in the transmission of the so-called *slidsjuka*, or sheath disease, due to the partially stuck ears in the leaf sheaths. Overall, there were differing opinions on the cause, but it was consistently observed that the damage occurred particularly in dry and hot years [115]. Field prevalence was relatively low in the 20th century, and thus, there are few descriptions of dwarfing symptoms in the scientific literature, but sometimes in the context of severe outbreaks in wheat [116,117,118,119,120]. *Slidsjuka,* or WDD, declined in Sweden around 1950 and occurred only sporadically in the following 30–40 years until the 1980s/1990s [121,122,123]. This decline was attributed to changes in agricultural practices. The abandonment of undersowing in winter wheat, which was common in the first half of the century, or even the increased use of combine harvesters, was considered to have had a positive effect on disease control [124].

The direct relationship between virus, vector, and symptoms was first reported in 1961 using samples from wheat fields in western parts of the Czechoslovakia [10,125]. However, there was still confusion about the cause, as no clear virus particles or possible pathogens could be detected [13]. The identification and current taxonomic classification of the virus did not occur until 1980, when, after three decades, there was again an increased incidence of the disease in a number of European countries [20]. In the late 1980s, a new disease (pieds chétifs) occurred in central France, causing severe damage in wheat, with yield losses of more than 50%, and was associated with a high incidence of the leafhopper *P. alienus* [126]. Initially, only Mycoplasma-like organisms were diagnosed in this context [117]. In collaboration with a Swedish research group, the disease-causing pathogen was identified as WDV [127].

From this time on, the occurrence of vectors and viruses was studied, with WDV occurring mainly in central France and adjacent areas but not in the coastal regions and south of the country [128,129]. The level of knowledge at that time was very low and was mainly based on studies from the Czech Republic [10], Sweden [20], and France [130]. In Germany, the first record probably occurred in 1990 near Dresden by Vacke [92] (Figure 2).

A concrete dispersal route cannot be deduced from the data. However, based on the biology of the animals and their activity, a natural spread over land seems most likely. The virus has been detected in the main Eurasian cereal-growing areas and in its region of origin in the Middle East. This can possibly be attributed to the fact that the climatic requirements for wheat cultivation, for example, match with those of *P. alienus*. Exceptions like India, as well as Canada and Australia, underline these theories.

The reason for the increasing spread of WDV and the increased occurrence in areas where WDV has been previously reported is not clearly understood but is probably caused by changes in agricultural practices. One of the main causes is assumed to be the increased use of ploughless tillage. Also, the EU regulation on the use of a large part of stubble fields after winter wheat cultivation as set-aside areas was thought to be favorable for *P. alienus* reproduction and overwintering. Avoiding set-aside areas after the occurrence of WDV-infected wheat and avoiding undersowing crops were therefore considered as possible control measures in Sweden [121]. Furthermore, harvesting with short stubble, early tillage in autumn, and avoiding early sowing had a positive effect on reducing the population of *P. alienus* [121]. Global climate change may also play a role in promoting the spread of vector-borne diseases. In this context, higher temperatures may favor the colonization of new habitats and hosts. Field monitoring is therefore essential, especially in cereal-growing regions, to identify additional regions where *P. alienus* may spread together with WDV [116,117,118,119,120] since the spread of WDV results from the migration of virulent vectors from wild or cultivated reservoirs into cereal fields [121,141]. Table 1 provides an overview of the history of WDD.

### 3.2. Host Range

The host range of WDV includes mainly monocotyledonous plants [37,142]. In addition to a variety of members of the Poaceae family, including important cereals such as wheat (*Triticum aestivum* L.), barley (*Hordeum vulgare* L.), rye (*Secale cereale* L.), oats (*Avena sativa*), and triticale [11,13,143], WDV also infects various wild and cultivated grasses, including *Bromus secalinus* L., *Lolium multiflorum* Lam. [13], *Avena fatua* L., *B. inermis Leyss*., *B. tectorum* L., *H. murinum* L., *L. perenne* L., *L. temulentum* L. [144], *A. sterilis* L., *A. strigosa Schreb*., *Poa annua* L. [103], *L. remotum Schrk.*, *Lagurus ovatus* L. [145], and *Apera spica-venti* (L.) *P. beauv*. [144], which are considered virus reservoirs [13].

### 3.3. Symptoms of WDD

The name of the virus is derived from its main characteristics, the disruption of the shoot growth and the formation of numerous shoots in wheat, resulting in the typical dwarf and bushy growth (Figure 3).

Furthermore, symptoms of WDV infection in wheat also include chlorosis, reduced root size, intense yellow or red discoloration of leaves with or without a mosaic pattern, deformation of leaves, reduced growth hardiness, delayed ear emergence, reduced number of ears as well as sterile flowers, significant yield losses and even complete plant death during early developmental stages of winter wheat and winter barley in winter and spring [13,121,146,147,148,149]. These are partly due to the side effects of infection, such as the effects of expression of viral suppressors of RNA silencing. Symptoms may also affect plant defense responses, leading to plant overreaction in the form of necrosis [150], chlorotic spots, and demarcated streaks on the leaves. The symptoms themselves first appear on the youngest and later on older leaves in association with small cracks and deformations on the youngest leaf, which are characteristic of the infection. This is followed by yellowing of the leaves at the leaf tips and margins with possible partial red coloration [13]. Symptomatic plants usually appear in patches in the field [11,13,148].

In addition to the described symptoms in wheat, the intensity of symptom expression varies among the other infested species. Symptoms in winter barley are similar to those of winter wheat, with no red coloration. Spring barley responds with a lower degree of dwarfing and yellowing of the leaf tips. Similar symptoms occur in winter rye, often associated with anthocyanin formation in leaves and culms. Spring rye shows only minor developmental depression, few leaf spots, and no disruption of generative plants. Oats show minor developmental depression, yellowing, and light red coloration [13]. Triticale shows no increased tillering after WDV infection compared to control plants, but spike-bearing culms shorten by half [151]. In *A. spica*, growth reductions of 20%, severe tillering, yellowing, and chlorotic spots were observed [13,103]. The wild grass *Poa annua* shows no symptoms after infection, while *Lolium perenne* and *Lolium multiflorum* showed tolerance to WDV in studies with longer plant viability after infection [11].

The extent of damage and the development of symptoms depends on the time of infection. Early infections of winter cereals at the 2–3 leaf stage during fall result in reduced winter hardiness, as well as severe developmental disorders, with pronounced symptoms and negative effects on yield as a result of ear formation that is often partially stuck in the leaf sheaths. The quality of the grains is reduced as they are dried out, shriveled, and partially unable to germinate [13,91]. The root system is also affected by WDV infection. As a result of the infection, there is a reduced formation of secondary roots. The roots appear shorter and thinner overall [91].

Infections in spring result in shortening of internodes and, in some cases, ears. In spring wheat, no severe developmental disorders but shortening of shoots could be observed when infestation occurred from the beginning of shooting to ear swelling (BBCH 31–45). Usually, the first signs of disease in winter wheat appear 18–25 days after infection. In general, symptoms in early-sown wheat are considered to usually appear four to six weeks after infection, while in late-sown wheat, the corresponding symptoms do not become visible until spring, provided the plants are able to overwinter. If infection occurs in spring or early summer, the incubation period lasts three to four weeks. In spring wheat, under greenhouse conditions, the first symptoms are expected 10–15 days after infection, while infections in the field have an incubation period of three weeks [13].

Symptoms caused by infection with *Barley yellow dwarf virus* (BYDV), which belongs to the Luteoviridae family and is transmitted by aphids, are visually similar to those caused by WDV. When infected in early fall, it causes WDV-like growth depression. The two viruses can only be distinguished from each other by double antibody sandwich enzyme-linked immunosorbent assay (DAS-ELISA) or polymerase chain reaction (PCR), so prior to the discovery of WDV, plants were probably often assigned to BYDV on the basis of dwarfism [151,152].

## 4. WDV and Its Vector

### 4.1. Taxonomy and Virus Transmission of P. alienus

WDV is transmitted by the leafhopper species *P. alienus*, which belongs to the class Insecta order Hemiptera, and uborder Cicadomorpha in the family Cicadellidae. The vector itself is a holarctic species that is common in grasslands and croplands [153]. Occurrence may be particularly high in fallow areas with many self-seeding plants of the Poaceae family. These may serve as reservoirs for WDV [12].

Many species of the Cicadellidae family are vectors of phytopathogenic viruses, including geminiviruses, phytorhabdoviruses, reoviruses, and marafiviruses [154,155,156,157]. In addition to WDV, *P. alienus* can persistently transmit a rhabdovirus, *Wheat Yellow Striate Virus* (WYSV, Nucleorhabdovirus genus) [158,159]. Furthermore, *P. alienus* appears to harbor entomopathogenic viruses that naturally infect insects and can only self-replicate in insect cells. In this context, filovirus-like particles were detected by Lundsgaard [160] in electron microscopic studies, which were confirmed as *Taastrup virus* (TV) and tentatively assigned to the Mononegavirales [160,161]. Using a next-generation sequencing approach, additional insect-specific viruses were detected, including *P. alienus iflavirus1* (PaIV1, genus Iflavirus, family Iflaviridae) [162], *Tàiyuán leafhopper virus* (TYLeV, genus Mivirus, family Chuviridae) [163], and *Hancheng leafhopper Mivirus* (HCLeV, genus Mivirus, family Chuviridae) [164]. Transmission electron microscopy (TEM) studies of WYSV-containing sites in salivary glands revealed the presence of reoviruses [165]. Reoviruses include insect-transmitted fijiviruses, which are the most common viral agents of a variety of diseases in gramineae, including *Fiji virus* (FDV) [166], *garlic dwarf virus* (GDV) [167], *maize dwarf virus* (MRDV) [168], *Mal de Rio Cuarto virus* (MRCV) [169], *oat sterility dwarf virus* (OSDV) [170], *Pangola stunt virus* (PaSV) [171], *rice black-streaked dwarf virus* (RBSDV) [26], and *southern rice black-streaked dwarf virus* (SRBSDV) [172]. Furthermore, the brown leafhopper *Nilaparvata lugens* has been found to harbor *Nilaparvata lugens reovirus* (NLRV), a fijivirus that exclusively infects insects [173]. Most published data suggest that *P. alienus* is the sole vector of WDV. Some authors have also described a transmission by *P. provincialis* [137,174]. However, due to the complex taxonomy of species belonging to the genus and the difficulties to distinguish individuals based upon morphological characteristics, the leafhoppers used within the studies are often poorly characterized. This could lead to contradictory results, especially regarding the role of species in WDV transmission [175,176,177].

### 4.2. Morphology of P. alienus

To easily differentiate adult *P. alienus* from other leafhoppers, several criteria related to the morphological characteristics of the insects’ head, abdomen, and wings can be used [178,179]. A characteristic of adult *P. alienus* is their brown coloration with transparent wings, which are longer than the abdomen with a length of 2.7–3.7 mm [180,181]. Accurate species classification requires the morphological description of the male genitalia due to the high variability of the morphological characteristics of the aedeagus. Identification of nymphs and females based on morphological characteristics is currently not possible. This approach often turns out to be unreliable [182,183].

The accuracy of identification of individuals could be improved by using several criteria in parallel, e.g., morphometric parameters in combination with other approaches, such as the emission of species- and sex-specific vibrational signals [184,185,186,187]. Only a few publications have described the vibrational signals emitted by leafhoppers during their sexual communication [182,186,188,189], and a combination of body and aedeagus characteristics combined with the analysis of vibration signals revealed geographic differences between species related to these characteristics. However, this may not only allow the identification of this species but also its origin. Therefore, future studies should include individuals from different countries to improve morphometric data [189]. A more straightforward approach that requires less expert knowledge is the use of DNA barcoding based on sequencing of the mitochondrial cytochrome oxidase I (COI) [190,191]. To date, phylogenetic analysis using DNA barcoding has only been performed for a limited number of species and individuals from Canada, Japan, and Korea [192,193]. Individual specimens of *P. confinis* and *P. helvolus* have already been found syntopic to *P. alienus* using this method [194].

### 4.3. Life Cycle of P. alienus

The life cycle of *P. alienus* has been well studied (Figure 4). High population densities can occur in September, making this the most critical period for WDV infections on young winter cereal plants. Extensive primary infections could be observed until December [12].

Embryonic development is influenced by environmental conditions like temperature and day length. Low temperatures in winter are necessary for the abolition of dormancy (termination) [196]. *P. alienus* shows seven embryonic stages with a total developmental duration of 16 to 24 days [12]. Depending on temperature, the first larvae hatch in early May. In this context, protandry can be observed, where males hatch earlier than females [196]. The wingless nymphs develop into male and female adult leafhoppers in five stages with a developmental duration of 26–39 days until early summer. Development duration varies, again depending on temperature, but also on host plant species and sex of the leafhoppers. In winter barley, 31 days can be assumed at a temperature of 20 °C [12,195]. After hatching, nymphs move through stocks exclusively by jumping, with older individuals being more mobile than the first two nymphal stages [197]. The newly hatched nymphs acquire the virus from host plants previously infected in the fall, which can lead to secondary infection of plants. It has been observed that the first imagines appear at the end of May, when the temperature sum of all days above 9 °C, measured from the 1st of January of a year, generally reaches 154 °C [12,198]. Fertilization and oviposition occur after the tenth day of the adult stage, so that the first generation begins oviposition in June/July, and the second-generation hatches about 18–20 days later [12,195] and lays its first eggs in early/mid-August. The duration of the entire egg-to-egg life cycle is 58 days [12,195], but higher temperatures may reduce this period, as demonstrated for *D. maidis* [199]. Dormancy egg laying is induced with the onset of a short day in mid/late August with a rate of 2–20%. From September onwards, up to 100% of eggs are laid as dormancy eggs [196]. Asexual reproduction, as observed in aphids, does not occur in leafhoppers [12,195].

In temperate climate zone, two to four generations per year have been observed so far, depending on environmental conditions [101,177,181], with four complete generations from spring to fall in cereal-growing regions of France, whereas only two *P. alienus* generations per year occur in northern Europe and northwestern China [200]. Population dynamics studies showed that the density of individuals can reach 43 adults/m^2^ [12]. The sex ratio in an adult population of *P. alienus* is close to 1 [200]. The number of adults decreases above a temperature of 10 °C [153]. Freezing temperatures of −5 °C leads to induced death of animals [12,198]. Temperatures above 35 °C have been associated with increased mortality [201]. In contrast, activity and population size of *P. alienus* increase significantly above a temperature of 15 °C. Thus, a very mild fall therefore leads to very active leafhoppers associated with increased WDV infection rates in the following summer [202].

### 4.4. Process of Virus Transmission

According to taxonomic affiliation [203] and based on electron microscopic observations [204], *P. alienus* belongs to the salivary sheath feeders (Auchenorrhyncha), which also includes most of the Sternorrhyncha (aphids, scale insects, psyllids). A salivary sheath is formed in the apoplast by secretions of gel saliva and surrounds the stylet as it moves through plant tissues toward the sieve elements, as shown in aphids [205]. When the stylet reaches the xylem or phloem, the uptake of sap from the vascular cells occurs for the extraction of nutrients [203]. Direct damage by *P. alienus* caused by sucking activity is considered less important than indirect damage caused by transmission of phloem-restricted WDV [13]. WDV is persistently, circulatively, and non-propagatively transmitted from plant to plant [101,174]. Mechanical, soil- or seed-dependent transmission has not been reported so far [20].

The characteristic of persistent transmission is that a single virus uptake by the vector is sufficient to transmit the virus for months after a short latency period, i.e., the time between the uptake of virus particles and the subsequent release via the salivary glands [20]. A latency period of one to several days is assumed [101,206,207]. Seventeen days after virus acquisition, transmission efficiency was found to be 90%. Transmission efficiency is influenced by environmental conditions, such as temperature, while transmission success depends on the virulence of the virus isolate and the susceptibility of the host [208]. Vector studies on *P. alienus* are currently focusing on evaluating the transmission of WDV, determining the host plant range, and observing probing behavior on a variety of plants [209].

To date, two pathways of virus movement within the vector and transmission to healthy plants are known. Similar to the persistent virus transmission of other insects, the virus can enter the salivary glands through the anterior midgut and hemocoel [210] or migrate into the lumen of the filtering chamber and on to the midgut lumen after entering the esophagus. Ten minutes after the first feeding, the virus is found throughout the midgut of the insect, and within the next ten minutes, it accumulates throughout the entire filter chamber, midgut, hemocoel, and salivary gland. Four hours after the first feeding, it is no longer detectable in the filter chamber, but it has accumulated in the midgut, hemocoel, and salivary glands, where it remains for the rest of the leafhopper’s life without replicating [207]. The transient direct transfer of particles to the salivary glands occurs within a few minutes, after which the normal circular, non-propagative pathway occurs with the recruitment of the anterior and midgut organs of the leafhopper [211]. Here, the WDV CP not only has an encapsulation function but is also involved in the retention and transmission of WDV in the leafhopper, virus propagation within the plant, and interaction with the Rep protein [105]. Once the vector has acquired the virus by ingestion [10,11,148], the virulent leafhopper can transmit the particles to new hosts each time it sucks. In this process, the virus particles are not lost during molting, so the virus remains in the vector for life. There, it interacts directly with the insect’s organs but does not replicate within the vector [148]. Although the WDV pathosystem is poorly documented in the literature, it has been clearly demonstrated that the virus is not transmitted vertically from virulent females to eggs. Vacke [10] assumed that after the acquisition, all developmental stages are capable of transmitting WDV. This was confirmed by Mehner et al. [11] using transmission tests with larval stages. Larval stages IV and V were more inefficient (22% and 9%, respectively) in terms of virus uptake compared to earlier larval stages and imagines (LI 43%, LII 50%, L3 45%, imago 41%) [11]. Larval stages appear to be more important than adult leafhoppers for WDV dispersal in this regard. Even at low densities, adults and larvae can cause significant yield losses by transmitting the virus to numerous host plants [198]. In the presence of the aphid species *Rhopalosiphum padi*, a negative effect on larval development, lifespan, and fertility of *P. alienus* has been observed. Studies of their interaction have ruled out food deprivation as a possible cause. It is hypothesized that the presence of aphids alters leafhopper behavior. This leads to an increase in the number of plants visited by individuals. Thus, this antagonistic interaction between aphids and leafhoppers, commonly found together in cereal fields, indirectly promotes the efficient spread of WDV [198]. Within an experimental approach, the highest infection rates were observed at temperatures of 25 °C. At higher temperatures, leafhoppers tended to settle on the ground, resulting in lower feeding rates and, thus, a decrease in transmission rates [201].

### 4.5. Host Range and Wild Reservoirs

In particular, the presence of wild grasses in stubble fields as virus reservoirs can lead to an extension of the virus infection period in autumn and promote the occurrence of the disease in spring [12]. The role of wild grasses as WDV virus reservoirs in cropland was demonstrated by Yazdkhasti et al. [212]. The results showed the potential role of ryegrass in the epidemiology of WDV [121] as a symptomless reservoir and underlined the wide host range of WDV [212]. In addition, removal of the overgrowth by plowing immediately after harvest is strongly associated with a reduction in leafhopper [12], probably reducing the spread of WDV from wheat and barley to wild grasses. The host range of *P. alienus,* as a first-degree oligophagous species [213], is mainly restricted to known host plants of the *Poaceae* [180,214]. Therefore, in experimental studies, *P. alienus* has always been reared on grasses such as *Hordeum vulgare* L. [11,195], *Triticum* spp. [198], and *Festuca gigantea* (L.) *Vill*. [160]. Data from field studies also indicate feeding on other plant species, including alfalfa, carrot [215,216], and ragwort [*Ambrosia artemisiifolia* L. (Asteraceae)] [217]. This indicates a possible diet of dicotyledonous plants and explains the detection of phytoplasma strains in the body of *P. alienus* [216,218,219]. These observations contradict the results of a previous study in which *P. alienus* was not able to survive longer than two days on the two non-grass plants, *A. artemisiifolia* and *Carex tomentosa* L. (Cyperaceae). However, in this study, the average survival time of the two species was longer than the starvation control. This is due to the ability of the leafhoppers to possibly take up xylem cell sap from non-host plants [220], where the nutrient and water uptake may contribute to increased survival [209].

### 4.6. Studies of Insect-Plant Interactions

The behavioral sequence for host plant acceptance of hemipteran insects starts after landing with an exploration of the plant surface, where the plant surface is scanned with the tip of the labium, followed by probing, including cell sap sampling [203,221]. For *Cicadellidae*, as observed in other hemipteran groups (e.g., aphids), probing seems to be critical to distinguish between host and non-host plants [222]. As a result, not every plant is accepted as a suitable host, and rejection may occur during various stages of probing on the way to the phloem [223,224]. To better understand the behavior of piercing-sucking plant pests and the mechanism of pathogen acquisition and transmission, electrical penetration graph (EPG) technique has been developed to provide real-time observation of the feeding behavior [225,226,227,228]. EPG is probably the most important and widely used technique for studying insect–host–plant interactions, pathogen transmission and acquisition, insecticide effects, and plant resistance [229,230,231,232,233,234]. Within an EPG measurement, insects and plants become integrated into an electrical circuit. The insect closes the electrical circuit by penetrating the plant with its stylet, acting like a switch. Insects and plants act as variable resistors, and different behavior patterns, as well as the stylet’s surrounding environment, affect the electrical resistance, leading to voltage fluctuations that result in different EPG waveforms representing different feeding behavior patterns [228,235,236,237]. The EPG method has been used, for instance, in studies on aphids [226,228,238], leafhoppers [239,240,241,242], mealybugs [243], phylloxerids [244,245], thrips [246,247], and whiteflies [248]. However, data on EPG studies of *P. alienus* are relatively limited in this regard [221,249,250,251]. Tholt et al. [209] suggested that viruses like WDV are transmitted between insects and plants during the EPG phase Ps4, where the stylet of *P. alienus* is located in the phloem’s companion cells and sieve cells. In this context, phase Ps4 can be further divided into phase 4a, similar to waveforms E1 shown by aphids, and is associated with the secretion of watery saliva into sieve elements, accompanied by virus transmission. Phase 4b appears to be a homolog to waveform E2 observed in aphids, indicating the ingestion of sieve element sap [209,238,252], probably accompanied by virus acquisition [209]. In addition, Ps4a resembled the X-wave that occurs in other leafhoppers [253,254]. Thus, phase Ps4 is particularly important for WDV transmission [209] and could be used during WDV resistance research.

## 5. Management of WDD, Its Vector and Virus

Knowledge regarding how to influence the population of *P. alienus* through appropriate countermeasures is currently insufficient. In field trials, parasitization has been observed very rarely [196]. In Italy, the parasitization of *P. alienus* larvae and imagines by *Gonatopus clavipes Thunberg*, *G. lunatus Klug* (*Heminoptera*: *Dryinidae* (cicada wasps)), and representatives of the family *Pipunculidae* (Diptera: eye flies) native to this country has been observed more frequently [195]. Predominantly in the first generation in May to June, larvae of *Gontopus sepoides Westwood* have been found on the abdomens of leafhoppers, acting as exoparasites, while *Alloneura nigritula Zetterstedt* (Pipunculidae) is more commonly found in October to November on *P. alienus* [255]. In addition, experiments have shown that *P. alienus* is preyed on by the spider *Tibellus oblongus* [256].

The actual lack of systematically evaluated, commercially available WDV-resistant and tolerant elite cultivars of wheat and barley means that protection of these cereals against WDV infection relies mainly on agronomic measures and the use of chemically synthesized control agents (insecticides) against *P. alienus.*

Prevailing cropping practices influence the presence and spread of plant virus diseases, closely correlating with the fluctuating incidence of WDD and the extent of yield losses. The timing of sowing, coordinated with the migration of vectors between fields, is a critical element of an integrated pest management (IPM) strategy [257]. The presence of infected reservoirs, e.g., wild grasses, leads to an increase in the incidence of many viruses, including MSV and WDV [143,258], which in turn involves the field hygiene aspect to reduce WDV infection. Another risk is irregular germination of seedlings [177], as *P. alienus* is attracted to patchy stands [17]. In addition, feeding behavior, population density, and activity, the latter influenced by weather conditions, affect the intensity and frequency of a WDV infestation [177,259]. A WDV infection is possible at different stages of development (Figure 2), with economic damage decreasing with later infection [17], as has been described for other viruses such as BYDV [260]. Furthermore, it has been shown in wheat that plant resistance can develop after the stage of pseudo stem break (Z30) at the time of the first node (Z31) [202].

Although IPM aims to reduce the application of chemically synthesized insecticides and other pesticides, it does not exclude the possibility of insecticide application. With regard to virus spread, the insecticide-induced reduction of vector insects has been shown to reduce the spread of insect-transmitted viruses [177,258,261]. However, the application of insecticides is associated with negative environmental side effects [262,263], including harmful effects on beneficial insects [264,265,266]. Together, the consideration of these aspects, as well as the broad public request and political will to reduce the use of insecticides, means that the focus for controlling WDV is mainly on agronomic measures and the breeding of resistant/tolerant varieties.

## 6. Resistance Research and Status Quo in Wheat

Abiotic and biotic factors exert a constant influence on plant populations. Naturally, plants have inherent defense mechanisms that make them resistant to virus invasion [267]. One way is to combat the virus by induced mechanisms, such as RNA silencing with small interfering RNA (siRNA) in response to the virus’s double-stranded RNA (dsRNA), hypersensitive response (HR), or nucleic acid methylation before infection occurs [268]. To date, nothing has been reported on effective and protective defence responses against WDV [269].

In recent decades, various studies have attempted to identify WDV-resistant germplasm among the available wheat and barley accessions. Disease resistance genes in wild relatives of wheat can serve as valuable sources for resistance breeding [270]. Differential resistance to *Soil-borne wheat mosaic virus* (SBWMV) has been demonstrated in *Ae. tauschii* and *T. monococcum* [271,272,273] and in *Ae. geniculata* to BYDV [274]. Furthermore, *Ae. caudata*, *Ae. ovata* and *Ae. triuncialis* have been shown to respond to WDV infection with milder forms of symptoms compared to spring wheat [13].

Transmission of the virus to the genotypes to be tested has been carried out in previous studies using the natural vector *P. alienus* or agroinfections. Phenotyping of infected plants is possible under field [3,147,149], and near-field conditions [275,276], or in the greenhouse [275,276,277]. For field inoculation with virus-bearing leafhoppers, both natural and artificial inoculation can be used. In order to protect the crops from natural insect infestation and bird-induced damage, trials can be conducted under semi-field conditions within a gauze house [275,276].

Within phenotyping for resistance, various agronomic parameters may be of interest. Virus infections with WDV affect the performance and yield of infected plants compared to healthy plants. Here, the traits of plant height, number of ears per plant, grains per ear, grain yield per plant, and thousand kernel weight (TKW) per plant can serve as suitable indirect parameters for characterizing resistance [278]. Between tillering and sprouting (BBCH 23–30), as well as after harvest (BBCH 92), a comparative symptom assessment from 1 to 9 can be performed according to Scheurer et al. [279].

Serological and molecular techniques are available for the detection of WDV infection as well as for a precise assignment of isolates to the corresponding strain designations. For the verification of WDV infections in the field, direct virus detection, via ELISA [280] and PCR [101,281,282,283], has proven to be a reliable method [152,284]. Differentiation of the WDV strains in the host plants and vector samples can be made on the basis of the characteristics of viral compounds (capsid proteins, nucleic acids). Due to the high sequence similarity between the CP of the isolates, serological differentiation of these using polyclonal antisera is not possible [147], but the use of monoclonal antibodies has been reported [285]. Several established molecular methods are available for the identification of WDV strain-specific sequences, such as standard PCR [80,102], restriction fragment length polymorphism (RFLP) [286], rolling circle amplification restriction fragment length polymorphism [104], and isothermal recombinase polymerase amplification methods [287]. In addition, molecular-based quantification assays in the form of real-time PCR assays targeting a conserved region of the CP gene sequence and using a Taq-Man probe have been added to the list of detection methods [174].

So far, no highly resistant WDV bread wheat variety is known. However, tendencies to favor different wheat varieties [288] and differences in susceptibility have been found (Table 2).

Based on yield reduction, studies were conducted on winter wheat to identify tolerant groups [149]. These showed only minor quantitative differences between the tested host plants and reference genotypes [3,147]. Most genotypes were susceptible to WDV infection, and only a few genotypes could be classified as moderately resistant. Within screenings, the Czech winter wheat cultivars ‘Banquet’ and ‘Svitava’ showed reduced virus levels, with moderate susceptibility at a yield reduction of 87.3–93.1% [149]. Moderate yield reductions of 82.5–92.6% after WDV inoculation were shown by the Russian cultivars ‘Belocerkovskaya,’ ‘Kharkovskaya,’ ‘Mironovskaya 808’, ‘Yubileynaya’ and ‘Kawvale’ and the Slovak and Czech cultivars ‘Astella,’ ‘Boka,’ ‘Bruneta,’ ‘Bruta,’ ‘Ilona,’ ‘Ina,’ ‘Mona,’ ‘Regina,’ ‘Saskia,’ and ‘Senta’ [147]. The winter wheat varieties ‘Mv Dalma’ and ‘Mv Vekni’ from Martonvásár (Hungary) were described by Benkovics et al. [289] as the first partially resistant varieties. In leafhopper transmission tests, both cultivars were infected (53%) but showed milder symptoms and a 100–10,000 times lower virus titer than the susceptible reference host cultivars ‘Mv Emese’ and ‘Mv Regiment’ (100% infection) four weeks after infection. A difference in the survival rates of the leafhoppers could not be determined. It can, therefore, be assumed that the resistance mechanism of the cultivars is based on the movement or replication of the virus and not on insect feeding [289]. ‘Mv Dalma’ carries a homozygous 1AL.1RS, while ‘Mv Vekni’ carries a homozygous 1BL.1RS rye translocation and contains several stem, leaf, and yellow rust resistance genes derived from *Aegilops ventricose* (VPM-1, SR38, Lr37, YR17) [289,290,291].

To clarify the genetic basis of partial resistance in ‘MV Vekni,’ in a recent work, F2 populations based on a cross between the susceptible cultivar Regiment were inoculated in greenhouse experiments, and quantitative trait loci (QTL) analysis was performed. Significant QTL were found for the peak markers RFL_Contig6053_2072 and Kukri_rep_c95718_868 on chromosome 6A for virus extinction (LOD = 22.6), which explained a phenotypic variance of 38.4%. The significant deviation from the expected segregation ratio of 3r:1s observed in this work indicated that the resistance is primarily inherited monogenetically due to the action of one major gene eventually accompanied by additional minor QTL that could not be detected within the analysis. The hypothesis of coupling rye introgression with WDV resistance in Vekni could not be confirmed in this work. Within the main QTL interval, among others, a gene encoding protein kinase activity could be identified [292]. These are involved in various defense mechanisms against geminiviruses, leading to attenuation and reduction of infection [293]. Furthermore, genes associated with DNA-directed transcriptional regulation in *Triticum aestivum* have been found to act as viral defense modulators, influencing the host-dependent DNA replication cycle [51,292].

In a recent study [294], the changes in transcriptome profiles of the resistant wheat genotypes ‘Svitava’ and ‘Fengyou 3’ compared to the susceptible cultivar ‘Akteur’ were investigated after WDV infection. The study provides insights into the specific transcriptome profiles and pathways associated with resistance and susceptibility to WDV in wheat genotypes. RNA-Seq analysis revealed significantly different expressions of transcripts in response to WDV infection in ‘Akteur,’ ‘Fengyou 3’, and ‘Svitava’ genotypes. Gene ontology (GO) analysis showed that different biological processes, cellular components, and molecular functions were activated in the tested genotypes. The resistant genotype showed significant activation of biological processes compared to the susceptible genotype. Certain classes of genes were affected by WDV infection. For example, transport activity was suppressed [294], which could prevent virus movement and accumulation [295]. On the other hand, oxidoreductase and lyase activities were activated [294], which are involved in defense responses and limit virus accumulation [296]. The ‘Svitava’ genotype suppressed reductase protein classes and chaperones. The latter group includes heat shock proteins (HSP), which play a role in viral DNA/protein aggregation and viral reduction [297,298,299]. Suppression of reductase activity is associated with a reduction in reactive oxygen species (ROS) accumulation, which is associated with better adaptation to viral infections [300]. Analyses of GO and KEGG metabolic pathways revealed reprogramming of several transcripts in response to WDV infection, particularly in the carbohydrate, energy, lipid, nucleotide, amino acid, glycan, and vitamin metabolism. Secondary metabolic and photosynthetic pathways were induced in ‘Svitava.’ The susceptible genotype showed down-regulation of photosynthesis-related carbon fixation genes, which, in contrast, were induced in the resistant genotypes. Transcripts for the biosynthesis of other secondary metabolites were upregulated in ‘Svitava’ and downregulated in ‘Fengyou 3’ and ‘Akteur,’ possibly contributing to higher resistance through their antiviral properties [294,301]. Transcription factors (TFs), including AP2/ERF, bHLH, MYB, and WRKY families, were highly enriched under WDV infection [294]. These TFs are known to regulate plant responses to various biotic and abiotic stresses [302,303]. In particular, ERFs have been linked to plant immune responses and resistance to plant viruses [304].

In greenhouse experiments with 13 wild and five domesticated wheat taxa of different ploidy, accessions of the species *Aeg. tauschii*, *Aeg. cylindrical*, *Aeg. Searsii*, and *T. spelta* showed WDV tolerance. The accessions were initially strongly affected by symptoms 28 days after infection (dpi). Thereafter, there was a decline in symptoms with a relative increase in leaves and shoots at 112 dpi. Within the study, domesticated wheat cultivars did not always show more severe symptoms, but there was a differential impact of infection on growth traits and leaf chlorosis in wild and domesticated wheat cultivars [277]. This could be attributed to a slight RNA silencing suppressor activity of the WDV proteins Rep and RepA [62,305]. Both viral proteins, when expressed in infiltrated transgenic leaves of *Nicotiana benthamiana* with a green fluorescent protein (GFP) reporter gene, resulted in the inhibition of post-transcriptional gene silencing (PTGS) and RNA silencing of the GFP reporter gene [305].

Within another study, 500 wheat accessions were phenotyped for WDV resistance by artificial inoculation in gauze houses. The majority of accessions showed a strong impact of WDV infection with a wide range of reductions in plant height (3.6–100%), number of ears (0–100%), and yield (2.3–100%) [275]. In contrast to Nygren et al. [277], domesticated wheat varieties within the panel did not have a generally higher infection rate than wild wheat varieties and relatives [275]. The authors concluded that the genetic bottleneck that arose during evolution and domestication did not necessarily lead to higher WDV susceptibility but that these variations created by ancestral hybridization were compensated for. During the study, the partially resistant genotypes ‘MV Dalma’ and ‘MV Vekni’ were confirmed with an average infection rate of 34.5% and 21.5%, respectively, and weaker symptom expression compared to susceptible varieties. In addition, 19 other sources of WDV resistance with lower infection rates than ‘MV Vekni’ were identified, including di-, tetra-, and hexaploid genebank wheat accessions. Ten *T. aestivum*, two *T. vavilovii*, two *T.* sp. (genebank accessions with unknown subspecies), one *T. boeoticum*, one *T. macha*, one *Ae. geniculata*, one *Ae. Bicornis*, and one *Ae. longissima* accession had lower infection rates than ‘MV Vekni.’ The cultivar ‘Fisht’ proved to be another resistant cultivar with a low average number of infected plants (5.7%) and less severe virus symptoms (average scoring value 2.3, for symptom scoring see [275]) compared to the reference cultivars ‘Mv Dalma’ (34.5%, 5.9) and ‘Mv Vekni’ (21.5%, 4.6) and the susceptible ‘Mv Regiment’ (64.9%, 6.7) as well as ‘Mv Emese’ (68.1%, 6.9). Overall, the results indicated that there are natural sources of WDV resistance within the wheat gene pool. A subpanel was also used to identify QTL for WDV resistance in hexaploid wheat. The putative 35 QTL (FDR, α < 0.05) for partial WDV resistance for the traits relative plant height (relPH), relative yield (relYield), and relative thousand kernel weight (relTKW) are located on chromosomes 1B, 1D, 2B, 3A, 3B, 4A, 4B, 5A, 6A, 7A, and 7B. Among them, the most significant QTL were detected on chromosome 1B, especially six QTL explaining more than 10% of the phenotypic variance (LOD 5.0–8.7) and two highly significant yield-related QTL explaining 18.3% of the phylogenetic variance (LOD 5.0–8.7), which can be used to develop molecular markers in resistance breeding. The QTL identified here could be associated with genes encoding DNA template regulation of transcription, splicing mRNA by spliceosome, gene silencing by RNA, and protein kinase activity [275]. Genes responsible for the regulation of DNA template transcription may serve as modulators of viral defense, particularly with respect to controlling the host-dependent DNA replication cycle of WDV [51]. Previous research on RNA-mediated gene silencing has also demonstrated the ability of geminiviruses to trigger post-transcriptional gene silencing (PTGS) [306,307], such that viral dsRNA is degraded during the RNA splicing mechanism to small interfering RNAs (siRNAs) that align and degrade silencing complexes to sequence-specific mRNA [308]. Also involved in plant resistance to geminiviruses are protein kinase domains through phosphorylation of viral pathogenesis proteins. The viral protein ßC1 is phosphorylated by SNF1-related kinases, which has negative effects on RNA silencing suppressor function or labeling for degradation in the 26s proteosome. As a result, delayed/reduced viral infection may be observed [309]. Overall, the results suggest that other resistance genes are involved in defense against WDV.

Previous studies have shown that resistance to various viruses is localized to the D chromosome. For example, resistance to *Soil-borne Wheat Mosaic Virus* (SBWMV) is localized on chromosomes 4D and 5D, and the resistance gene encoding alleles on chromosome 5D is due to *Aegilops tauschii* [310,311]. Other highly significant marker-trait associations (MTA) were found on chromosome 2D for resistance to *Wheat spindle streak mosaic virus* (WSSMV) [312]. Of 35 QTL identified, 25 QTL, explaining between 7.4 and 18.3% of the phenotypic variance, were verified in four biparental populations with the cultivar ‘Fisht’ as a parent [275]. Within the segregation analysis, two of the markers showed significant effects on relYield, eleven on relTKW, and ten on relative virus titers. The QTL on chromosome 1B consistently showed highly significant effects in all four populations [275].

A recent QTL study revealed two additional highly significant QTL associated with WDV resistance [313]. The primary QTL, Qwdv.ifa-6A, mapped to the long arm of chromosome 6A between markers Tdurum_contig75700_441 (at 601,412,152 bp) and AX-95197581 (at 605,868,853 bp). Qwdv.ifa-6A originated from the Dutch experimental line SVP-72017 and showed a strong effect in all populations, explaining a significant proportion (up to 73.9%) of the phenotypic variance. The second QTL, Qwdv.ifa-1B, was located on chromosome 1B and derived from the susceptible parental line P1314. The QTL is possibly linked to the 1RS.1BL translocation, which originated from the CIMMYT line CM-82036. Qwdv.ifa-1B was responsible for a substantial portion (up to 15.8%) of the phenotypic variance in WDV resistance [313]. The efficacy of the rye chromatin segment 1RS.1BL against Wheat Streak Mosaic Virus (WSMV) has been reported previously [314], but there is no evidence to date that the same gene confers resistance to both WDV and WSMV. The QTL mapped on the short arm of chromosome 1B in the study by Pfrieme et al. [275] overlaps with the Qwdv.ifa-1B QTL identified within the study by Buerstmayr and Buerstmayr [313]. Although Fisht has the preferable allele on chromosome 1B, the presence of the translocation 1RS.1BL remains unclear. Thus, it remains uncertain whether ‘Fisht’ and P1314 (the resistance donor for Qwdv.ifa-1B) have the same resistance gene. This study has shown that Qwdv.ifa-6A and Qwdv.ifa-1B are clearly additive, suggesting that the pyramidization of resistance QTL could increase both the durability and extent of resistance [313].plants-12-03633-t002_Table 2Table 2Overview of the key findings of WDV resistance breeding in historical sequence.TimeEventReference1982Report: WDV shows tendencies to prefer different wheat varieties.[288]2000Screening: Description of five Russian varieties as well as ten Slovakian and Czech varieties with moderate yield reduction after WDV infection.[147]2005Screening: Description of the Czech winter wheat varieties ‘Banquet’ and ‘Svitava’ with reduced virus titer, moderate susceptibility, and yield reduction.[148]2010Screening: Description of the Hungarian winter wheat varieties ‘Mv Dalma’ and ‘Mv Vekni’) as partially resistant varieties.[289]2015Screening: Proof of WDV tolerance of accessions of the species *Aeg. Tauschii*, *Aeg. Cylindrical*, *Aeg. Searsii*, and *T. spelta*.[277]2022Screening: Identification of 19 sources of WDV resistance with lower infection rates than ‘MV Vekni,’ including di-, tetra-, and hexaploid genebank wheat varieties as well as the winter wheat variety ‘Fisht.’[275]2022Genome-wide association study: Detection of 35 putative QTL for partial WDV resistance on chromosomes 1B, 1D, 2B, 3A, 3B, 4A, 4B, 5A, 6A, 7A, and 7B.[275]2022QTL analysis: Identification of two significant QTL on chromosome 6A in the variety ‘Mv Vekni.’[292]2023Transcriptome analysis: A study of changes in resistant wheat genotypes ‘Svitava’ and ‘Fengyou 3’ compared to susceptible cultivar ‘Akteur’ after WDV infection.[294]2023QTL study: Identification of a QTL on chromosome 6A in the Dutch experimental line SVP-75360 and a QTL on chromosome 1B of line P1361.[314]2024QTL study: Identification of QTL in the winter wheat variety Fisht.


The utility of the discovered QTL for wheat breeding depends on their ability to predict quantitative WDV resistance in a range of genetic backgrounds. For breeding purposes, QTL associated with resistance should explain at least 10% of the phenotypic variance. Their pyramiding is an interesting approach to increase resistance to WDV [275,315,316,317], as already shown for BYDV in barley [278,318]. The use of the identified QTL in marker-assisted selection can be achieved by developing PCR-based markers from verified array-based markers. For example, the use of competitive allele-specific PCR markers (KASP) developed from flanking marker sequences offers an efficient approach in hexaploid wheat [319,320,321]. The introduction of WDV tolerance can be facilitated by the use of molecular markers, avoiding artificial inoculation with virus-bearing leafhoppers, which is difficult to integrate into applied breeding programs.

## 7. Conclusions

WDV is a worldwide virus disease that affects most cereals and grasses. As a result of climate change, the importance of insect-transmitted viruses will inevitably increase in the coming years. Research conducted within the last decades allows a description of the biology of the putative vector, the virus, and the plant hosts. In this context, the epidemiology of WDV is characterized by the presence of different strains, recombinants, and virus species, as well as a complex taxonomy of vectors and a contradictory host range. Although WDV as a DNA virus is thought to have a lower mutation rate compared to RNA viruses, putative new variants, and recombinants have already been detected in reservoirs and crop species in recent years. Since there are no approved chemical control agents in the European Union, agronomic measures are currently the only way to control WDV. The detection of the first WDV-resistant genotypes and QTL in wheat indicates that resistance is present in the cereal pool. As indicated by this review, further experimental studies on WDV resistance and the epidemiology of the vector are needed and promising, especially given the economic importance of this viral disease. The development of resistant cereal varieties offers the prospect of minimizing the spread and losses due to WDV infections.

## Figures and Tables

**Figure 1 plants-12-03633-f001:**
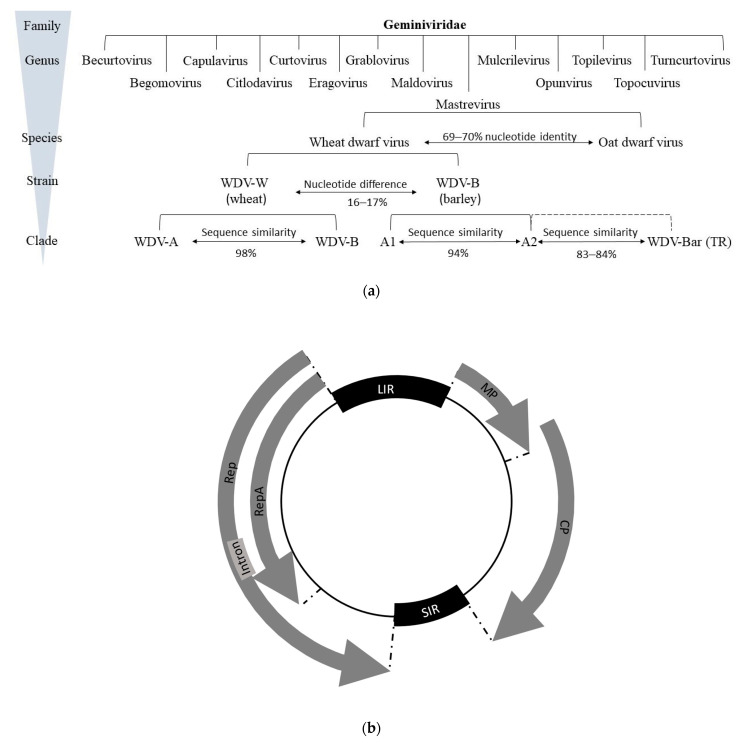
Classification and genomic organization of wheat dwarf virus (WDV): (**a**) classification of the family Geminiviridae is based on their molecular and biological characteristics. WDV species belong to the mastreviruses and consist of the main strains of wheat and barley, to which the various isolates are subordinated in clades. The percentage of nucleotide similarity is given for the species, strains, and clades. WDV Bar [TR] refers to the recombinant isolate between a barley isolate and a yet unknown member of the mastreviruses. (**b**) Genomic organization of mastreviruses, which include wheat dwarf virus (WDV). These have a circular ssDNA genome (black circle) and four ORFs. Code of viral proteins: MP—movement protein, CP—capsid protein, RepA—replication-associated protein, Rep—replication initiation protein. Also shown are the non-coding regions of the large intergenic region (LIR) and small intergenic region (SIR).

**Figure 2 plants-12-03633-f002:**
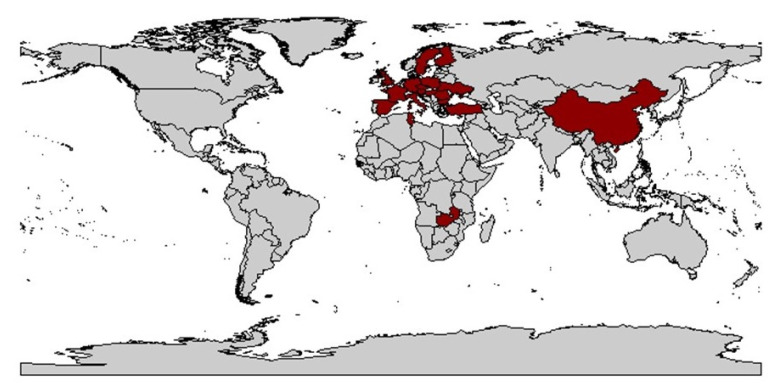
World map with countries where WDV could be detected (marked in red). WDV was reported in Ukraine [131], Romania [13], Bulgaria [131], Hungary [131], Italy [118], France [47], Sweden [20], Poland [132], Finland [18], Spain [133], the United Kingdom [108], Austria [108] and Slovenia [134], as well as regions in Iran [135], the Middle East (Turkey [109], Africa (Tunisia [120] and Zambia [136]), West Asia (Syria [137], and China [138,139]) [140].

**Figure 3 plants-12-03633-f003:**
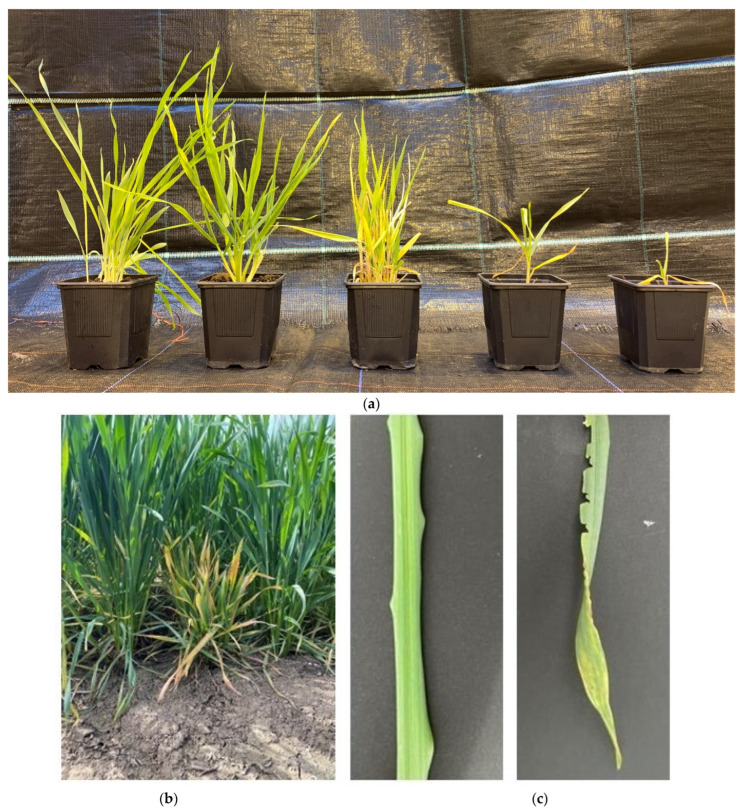
Eight-week-old wheat plants with different degrees (symptom scoring 1, 2, 5, 6, 8) of dwarfing in the greenhouse depending on their genotype (**a**) and at BBCH stage 30–39 in May 2021 under field conditions (**b**) after artificial inoculation with symptom-bearing in the middle of the image. (**c**) Leaves of WDV-infected plants (**left**) show a stripe-like lightening compared to healthy leaves (**right**), which later develops into yellowing.

**Figure 4 plants-12-03633-f004:**
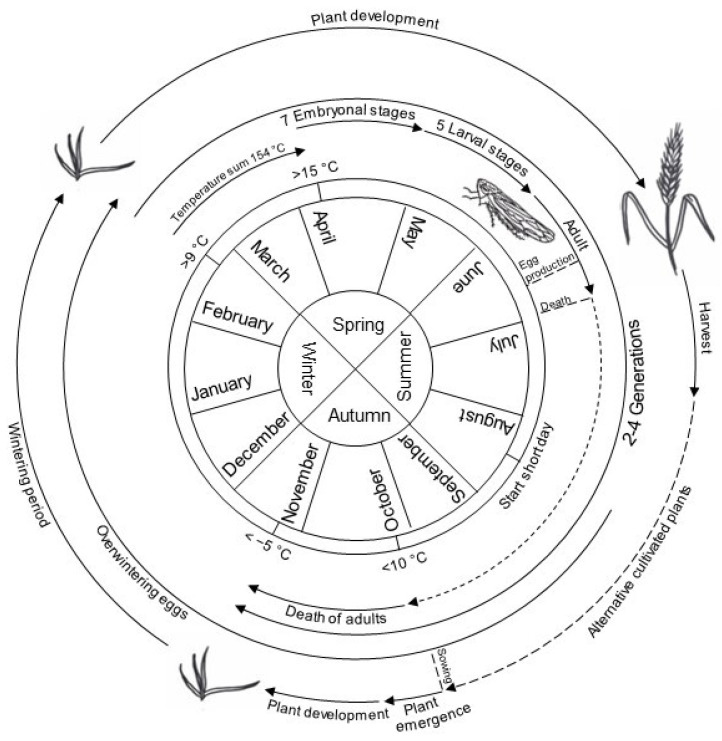
Schematic representation of the life cycle of winter cereals and *Psammotettix alienus*. The major developmental stages of host cereal plants (from sowing to harvest) are represented by the outer circle. The successive and overlapping biological cycles of *P. alienus* are represented by arrows in the inner circle. Under optimal conditions (20 °C, 70–95% relative humidity, 18/6 light/dark hours), the life cycle length (from egg to adult death) is 71 days [195]. Eggs produced in the fall overwinter on cereals and hatch in the following growing season (the next spring). According to Manurung et al. [12], the duration of the five larval stages (L1 to L5) is 5.9, 5.1, 5.6, 3, and 9.4 days, respectively. The seven-day-old adults can mate to produce the next generation of insects [12].

**Table 1 plants-12-03633-t001:** Overview of the historical development of WDV and its evidence in the individual countries in relation to its reference in the literature. For some events, no direct dates could be derived from the literature, so only a time span could be given.

Time	Event	Reference
Early 20th century	The first observed dwarfing symptoms of wheat, called *slidsjuka*	[114,115]
Early 20th century	Relatively low field prevalence of WDV; only a few symptoms of dwarfing have been described in scientific literature	[116,117,118,119,120]
Early 1950s	Less undersowing in wheat; increased use of combine harvesters	[124]
Around 1950	Decline of *slidsjuka* due to changes in agricultural practices	[121,122,123,124]
1950–1980/1990	*Slidsjuka* occurred sporadically	[121,122,123]
1961	The first report of a direct relationship between virus, vector, and symptoms; no virus particle detected	[10,125]
1980	Increased incidence of disease in European countries	[124]
1980	Identification and taxonomic classification of WDV	[124]
1981	Leafhopper *P. alienus* was made responsible for WDV occurence	[114]
Late 1980s	A new disease (pieds chétifs) occurred in France in association with *P. alienus*; the disease was identified as WDV	[126,127]

## Data Availability

Not applicable.

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
