# Peer review of "The Past, Present, and Future of Wheat Dwarf Virus Management—A Review"

_plants, 2023, doi:10.3390/plants12203633_

Round 1
Reviewer 1 Report
The present review manuscript entitled ‘The Past, Present and Future of Wheat dwarf virus Management- A Review’ presented in reasonable style. Theme looking good. However, authors are advised to read the manuscript carefully again and correct the language accordingly.
Abstract may be extended.
Add more tables by accumulation of past and present status.
English and Grammer also need the modification
Conclusion part must be added with separate subtitle.
References may be revise. If not very essential, please remove references published before recent past 10 years. Subsequently Add more relevant references published in reputed journals.

English and grammar may be improved
Reviewer 2 Report
"The Past, Present and Future of Wheat dwarf virus Management" is a well written review.
Please include a Figure with Phylogenetic Tree using the "Mastrevirus" sequences.
You may cite following reference on "Plant Resistance to Geminiviruses": https://biblio.iita.org/documents/S20InbkPatilPlantNothomDev.pdf-c1c85057a36d00c1bca8600d973d2cdc.pdf
Please check again for any grammatical errors. Here are some minor comments:
1. Line #24: “AD” should be in capitals: 8th century a.d.,
2. Line #164: virus has to pass through plasmodesmata.
Please check again for any grammatical errors. Here are some minor comments:
1. Line #24: “AD” should be in capitals: 8th century a.d.,
2. Line #164: virus has to pass through plasmodesmata.
Reviewer 3 Report
1. The abstract should be extended including the background of WDV, what you discussed and the perspectives 2. The subtitle of 5 and 6 should be improved because the authors only discussed the “Status quo of the resistance in wheat “ in the 6. Resistance Research part, I suggest just using one title is enough. 3. English need to be improved.
Minor editing of English language required
Round 2
Reviewer 2 Report
Accept